# Improving osteoarthritis care by digital means - Effects of a digital self-management program after 24- or 48-weeks of treatment

Leif E. Dahlberg[1,2], Andrea Dell'Isola🄳[1], L. Stefan Lohmander🄳[1,2], Håkan Nero🄳[1,2]*

**1** Department of Clinical Sciences Lund, Orthopaedics, Lund University, Lund, Sweden, **2** Arthro Therapeutics, Malmö, Sweden

* hakan.nero@med.lu.se

**Data Availability Statement:** Data from this registry contains potentially identifying and sensitive patient information and cannot be shared publicly. Anonymized data will be shared with

## Abstract

### Background

Osteoarthritis (OA) is highly prevalent in older adults and a growing cause of disability. Easily accessible first-line treatment of OA is increasingly important. Digital self-management programs have in recent years become available. Evidence of short-term effects of such programs are abundant, yet reports on long-term benefits and adherence to treatment are scarce. The current study's objective was to investigate the long-term pain and function outcomes of people with hip or knee OA participating in a digital self-management programme.

### Methods and findings

In this longitudinal cohort study, individuals with hip and knee OA, from the register of a digital self-management program and with 0-24-week (n = 499) or 0-48-week adherence (n = 138), were included. The treatment effect in terms of monthly pain (NRS, 0–10 worst to best) and physical function (30-second chair stand test (30CST), number of repetitions) change were investigated using a mixed model, controlling for the effect of age, body mass index (BMI), gender and index joint. For the 24-week sub-sample, pain NRS decreased monthly by -0.43 units (95% CI -0.51, -0.35, mean knee pain from 5.6 to 3.1, and hip pain from 5.9 to 3.8) and 30CST repetitions increased monthly by 0.76 repetitions (95% CI 0.64, 0.89 mean for knee from 10.0 to 14.3, and for hip from 10.9 to 14.8). For the 48-week sub-sample, pain decreased monthly by -0.39 units (95% CI -0.43, -0.36, mean knee pain from 5.7 to 3.2, and hip pain from 5.8 to 3.8), and repetitions increased by 0.72 repetitions (95% CI 0.65, 0.79, mean repetitions for knee from 10.3 to 14.4, and for hip from 11.1 to 14.9). There were no clinically relevant effects on the improvement of pain or function by any covariate (age, sex, index joint). The lack of a control group and randomization limit our ability to explain the mechanisms of the observed results.

### Conclusions

Continuously participating in a digital OA treatment program for 6 or 12 months was associated with a clinically important decrease in joint pain and increased physical function, in hip

researchers completing a data request with an approved research proposal and a data access agreement, signed by all parties. Contact data@jointacademy.com for more information.

**Funding:** Funding was received by LED from Vinnova - Sweden's Innovation Agency (grant number: 2016-04187, www.vinnova.se) and Stiftelsen för Bistånd åt Rörelsehindrade i Skåne (grant number: 2019-01-20, www.stiftbistandskane.se) to the Department of Clinical Sciences Lund, Orthopaedics, Lund University, Sweden. In kind support (data gathering and extraction) was provided by Arthro Therapeutics Inc. The funder provided support in the form of consulting fees for authors [LED, LSL, HN], but did not have any additional role in the study design, data collection and analysis, decision to publish, or preparation of the manuscript. The specific roles of these authors are articulated in the 'author contributions' section.

**Competing interests:** Joint Academy® is a product of Arthro Therapeutics® (AT), a Swedish e-health company. Håkan Nero and Stefan Lohmander are part-time consultants for AT, and Leif Dahlberg is CMO of AT. This does not alter our adherence to PLOS ONE policies on sharing data and materials. There are no other competing interests to report.

and knee OA. Future research should follow OA-related outcomes in participants that end their treatment to explore when and why that decision was made.

## Introduction

Osteoarthritis (OA) is the most common joint disease and one of the main causes of musculo-skeletal disability worldwide [1]. More than 10% of men and 18% of women aged older than 60 suffer from symptomatic OA [2]. Driven by the increasing longevity of the population combined with the rise of obesity, joint injuries, and physical inactivity, OA is now the fastest growing cause of disability worldwide [3–5].

National and international guidelines recommend education and exercise as first-line treatment for OA due to the effectiveness in reducing pain and disability [6–12]. However, reports show a low uptake of these recommendations with only 36% of patients receiving a joint replacement having been provided a first-line intervention, and less than 50% receiving appropriate care for their OA [13, 14]. Without immediate action, the rising OA prevalence will be a challenge without solution for the health care systems and for society worldwide [7, 15].

To tackle the increasing burden of OA and increase uptake of recommended first-line treatment, self-management programmes have been initiated in several countries [16–18]. Despite their success in improving patients' symptoms, the high prevalence of OA and difficulties related to their implementation, these programmes reach only a minority of the suffering OA population that would benefit from education and exercise [17, 19, 20].

Digital self-management programmes have been developed to further facilitate access to first-line treatment for OA and to aid patients in maintaining a long-term exercise regime [21, 22]. Early reports have shown a reduction in pain, disability and desire for surgical treatment in patients with OA after six weeks in a digital self-management programme [23, 24]. However, there is a dearth of knowledge surrounding the long-term effectiveness of such programmes, or adherence and effectiveness of the face-to-face counterparts [18, 19]. Hence, the main objective of the current study was to report on the long-term outcomes (24 and 48 weeks) of people with hip or knee OA participating in a digital self-management programme delivering first-line OA management, and statistically investigate the mean treatment effect of duration on pain and physical function, as well as differences in pain over time between hip and knee OA with or without additional covariates.

## Methods

This was an observational longitudinal cohort study approved by the regional ethics committee of Lund University and the Swedish Ethical Review Authority (Dnr: 2018/650 and 2019–02232). Written informed consent from participants was obtained during registration. The study adheres to the STROBE guidelines for observational studies [25] (S1 Checklist).

### Setting and participants

Participants joined the digital OA self-management and education programme (see Intervention below for details), through recommendation by their local orthopaedic surgeon or physiotherapist, and via online advertisements and campaigns placed on search engines and social networks. Included participants had a radiographic and or clinical diagnosis of hip or knee OA from a physical therapist or physician (95% of all patients at the date of data extraction). Individuals without a prior diagnosis had clinical OA confirmed by an orthopaedic surgeon or physiotherapist via telephone (diagnosis according to NICE criteria and Swedish National

Guidelines, and confirming the absence of any red flag symptoms), or if deemed necessary were recommended to seek face-to-face care before inclusion in the programme.

Data was extracted from the digital self-management programme registry on the 13th of March 2019. At that time point, the register contained data from 1709 Swedish participants (710 individuals had started ≥48 weeks before data extraction) that had reported one of their knees or hips as their most symptomatic joint (index joint), had been treated in the programme for at least three weeks with a minimum adherence of 70% and had registered ≥24 weeks before data extraction. Adherence was defined as the percentage of completed activities (exercises, text or video lessons on OA, and quizzes on lesson material) per the pre-defined period (the cut-off of 70% for the initial three weeks represents about 5 out of 7 days per week performing recommended activities), and mean adherence was defined as the group mean for the period of interest. Outcome analysis was made in two separate sub-samples; participants with a pain report at start and at week 24 or adjacent week (+- 4 weeks) and at start and at week 48 or adjacent week (+- 6 weeks), respectively. Adjacent weeks were added to collect the maximum amount of data (around 50% of active participants had reported pain at week 24 and 48, respectively). Hence, if a pain report was missing at week 24 or 48, the report from the closest week available was used (if two pain reports were available at the two most adjacent weeks, e.g. week 25 and 23, mean pain for these two was used). Those included in the 48-week sub-sample were excluded from the 24-week sub-sample, to enable comparison.

## Intervention

The intervention consisted of a digital, structured and individualized treatment programme for people with hip or knee OA (Joint Academy®; www.jointacademy.com). The programme consists of instructions for neuromuscular exercises appropriately adjusted to each patient in regard to degrees of complexity and difficulty. Exercises are distributed daily during the whole participation period, in general, two per day. While rating perceived difficulty and adding comments, patients also indicated when exercises were completed. Program continuation was halted until all exercises for the day were marked as completed. Information (based on current OA management guidelines and research) in the form of text or video lessons (with quizzes on the material after each episode) on subjects related to OA, OA symptoms and its management is also distributed to each participant. The lessons come packed in themes, with each theme containing 1–5 lessons where participants receive a theme per week the first six weeks, and then every other week, for a total of 70 lessons over a 48-week period. Completion of a lesson was indicated by the patient answering the quiz correctly. Additionally, continuous access to and dialogue with a physiotherapist through an encrypted chat function, and/or telephone, is provided.

## Outcome measures

In this program, several measures are collected at separate time points. There are weekly measures (joint pain), bi-weekly measures (physical function), and quarterly measures (other instruments not relevant for this study). Joint pain was assessed at baseline, and weeks 12, 24 and 48 using the Numerical Rating Scale (NRS, discrete boxes 0–10) with the instruction [26]; *Mark on this scale how much pain you had the last week in your hip/knee*, followed by a 0–10 scale where 0 was defined as *No pain* and 10 was defined as *Maximum pain*. Minimal clinically important change (MCIC) of pain was defined as an improvement of 20% (slightly or moderately important improvement according to Tubach et al., 2012) [27]. As a measure of physical function, the 30CST from week 12, 24 and 48 were used [28], performed by the participant with the help of an instruction video with a coupled visual timer. The patient entered the

performed number of repetitions after each test. Physical function data was handled similarly to the NRS, with week 24 ±4 adjacent weeks and week 48±6 adjacent weeks included for those with available NRS data. All outcomes were self-assessed and self-entered using the digital programme interface and chosen based on the International Consortium for Health Outcomes Measurement Standard Set for Hip & Knee Osteoarthritis (ICHOM) [29]. Data on the participants' overall health and characteristics (age, gender, BMI), as well as OA-related factors (most painful joint, afflicted side), were collected at inclusion.

## Statistical analysis

Summary data are described by the mean value, standard deviation and number of observations or the number and percent of the categories of interest. Comparisons of baseline data between the 24-week sub-sample and excluded participants (with missing data due to ending treatment or not reporting pain) were performed using independent samples t-test and the Fisher's Exact test (for dichotomous variables). The group-specific mean treatment effect of duration on pain and repetitions (30CST—physical function), as well as differences in pain over time between hip and knee OA with or without additional covariates, were estimated and tested using random slopes and intercepts models. Pain development over time was plotted for hip and knee OA, respectively.

To describe patients adhering for six months with contrasting pain severity at baseline, participants were divided into same-sized tertiles based on reported baseline pain, and mean pain per time point was calculated and plotted for each group (not performed for the 48-week sub-sample due to small numbers per group).

Significance level was set to $p < 0.05$, and p-values and 95% confidence intervals were reported when applicable. Statistical calculations were performed in SPSS Version 25 (IBM Corporation, New York, USA) and Stata 15.1 (StataCorp LLC, Texas, USA).

## Role of the funding source

The funding source had no role in any part of this study. The in-kind support sponsor, Arthro Therapeutics, aided in the collection and extraction of data from the registry, otherwise was not in any way involved in design, analysis or interpretation of data, writing of the text or submitting the paper for publication.

## Results

After identifying active users at week 24, a total of 920 individuals were found, whereof 290 individuals had not reported their pain into the register at or around week 24. Excluding those with 48-week data, 499 individuals reporting pain were included in the 24-week sub-sample. For the 48-week sub-sample, a total of 138 individuals (n = 7 missing 24-week data included) with pain data at 48 weeks were included (Fig 1).

## 24-week sub-sample

There were no differences between the 24-week sub-sample (n = 499) and excluded participants (combining those ending treatment, not reporting pain at 24 weeks or excluded due to having reported pain at 48 weeks also, n = 1210) in regard to age, BMI, baseline pain, baseline physical function, or the distribution of gender and index joint (for details, please see S1 Table).

Percentage of those reaching MCIC in pain after 24 weeks was 72% while mean adherence (SD) was 75% (18%). Descriptive data separated by index joint is reported in Table 1. Pain

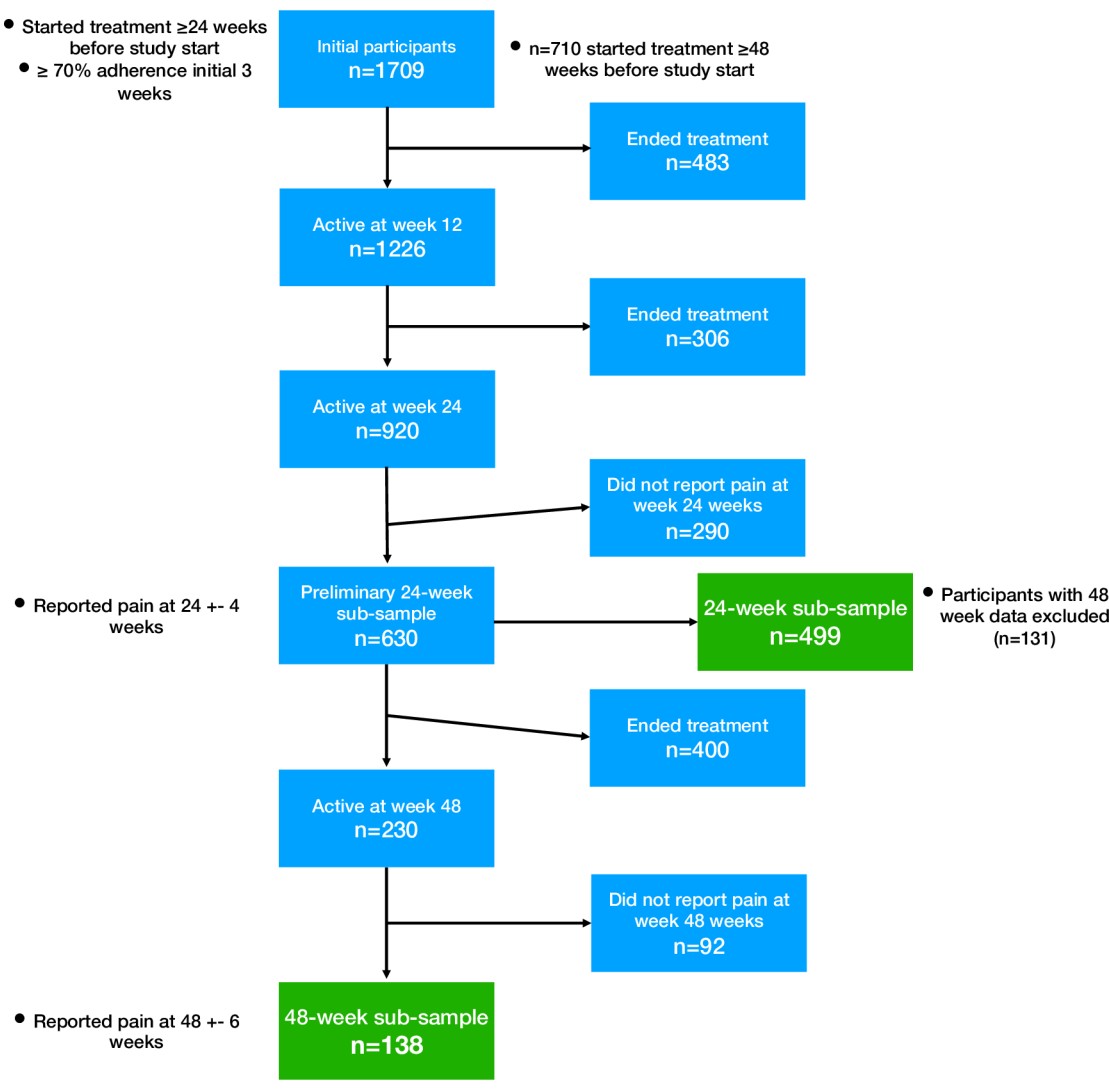

**Fig 1. Flowchart describing the disposition of patients.**

change over time per index joint showed continuous pain relief over 6 months of adherence for both joints (Fig 2). Visual inspection of plotted mean pain per pain severity group suggested a difference in absolute change, yet similar pain relief in terms of relative change for both hip and knee (Fig 3).

## 48-week sub-sample

Percentage of those in the 48-week sub-sample reaching MCIC in pain was 67% while mean adherence (SD) was 74% (21%). Descriptive data separated by index joint is reported in Table 2. Plotted pain change over time per index joint showed maximum pain relief achieved at around 6 months, and no signs of worsening pain for either joint, for up to 12 months (Fig 4).

**Table 1. Descriptive data for knee and hip OA in the 24-week sub-sample (n = 499).**

| Characteristic | Knee (n = 301) | Hip (n = 198) |
|---|---|---|
| Age, mean (SD) | 64 (9) | 63 (9) |
| Female, n (%) | 220 (73) | 152 (77) |
| BMI, mean (SD) | 28.2 (4.8) | 26.5 (4.9) |
| **Weight status, n (%)** | | |
| • Underweight (BMI <18.5) | 1 (0) | 1 (1) |
| • Normal weight (BMI 18.5–24.9) | 67 (22) | 83 (42) |
| • Overweight (BMI 25–29.9) | 141 (47) | 79 (40) |
| • Obese (BMI ≥30) | 92 (31) | 35 (18) |
| **Baseline pain per severity group** | | |
| • Low, mean (SD) | 3.3 (1.3) | 4.1 (0.9) |
| • Moderate, mean (SD) | 5.6 (0.5) | 6.4 (0.5) |
| • Severe, mean (SD) | 7.6 (0.8) | 7.7 (0.8) |

BMI = body mass index. NRS = Numeric rating scale. OA = osteoarthritis.

## Treatment effect over time

The treatment effect was modelled in terms of monthly pain change and monthly change in repetitions as a function of treatment duration for 24- and 48-week sub-samples using a mixed model with random slopes and intercepts. The models yielded marginally different estimates for pain with a decrease of -0.43 units (95% CI -0.51, -0.35) and -0.39 units (95% CI -0.43, -0.36) per month for the 24- and 48-week sub-sample, respectively. Similarly, physical function

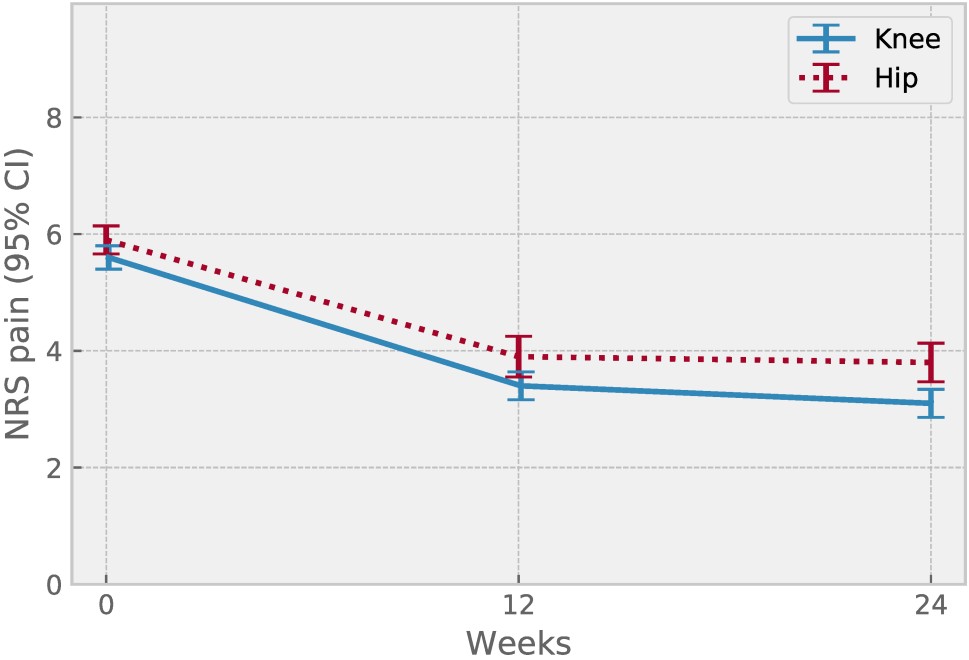

**Fig 2. Pain change over time, per index joint, for the 0 to 24-week subgroup of patients.** Symbols show mean pain value at 0, 12 and 24 weeks for patient subgroups with index joint knee (n = 301) or hip (n = 198). Error bars represent 95% confidence intervals.

## Pain over 24 weeks by baseline pain tertile

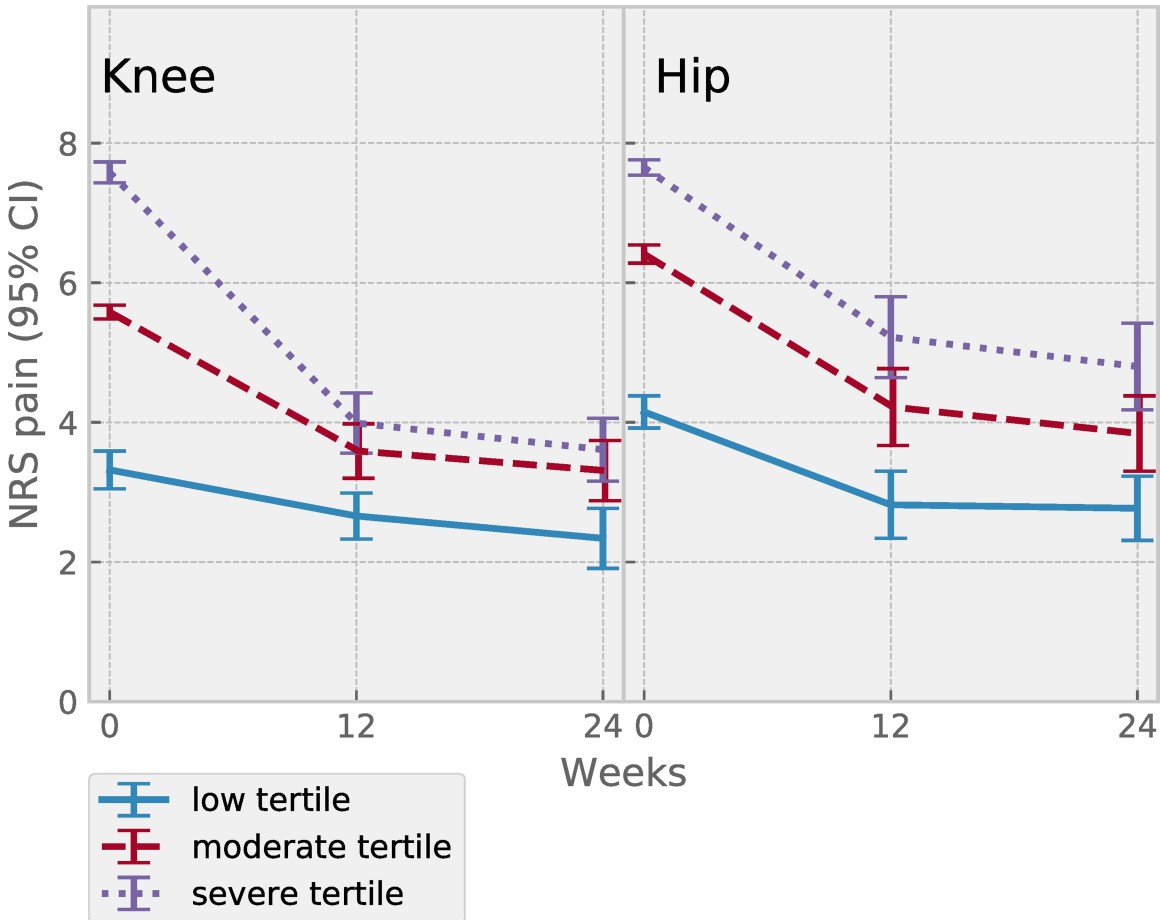

**Fig 3. Joint pain change over 24 weeks stratified in tertiles by baseline pain severity for the 0 to 24-week subgroup of patients.**
Symbols show mean pain value at 0, 12 and 24 weeks for patient subgroups with index joint knee (n for tertiles low 100, mid 99, high 102) or hip (n for tertiles each 66). Error bars represent 95% confidence intervals.

increased by 0.76 repetitions (95% CI 0.64, 0.89) and 0.72 repetitions (95% CI 0.65, 0.79) per month, for the 24- and 48-week sub-sample respectively. The group difference (comparing 24- and 48-week sub-samples) in pain change did not reach statistical significance.

BMI, index joint or gender did not importantly influence pain or physical function over time. Pain improvement was not affected by age, yet with increasing age, there was a statistically significant but not clinically relevant decrease in the improvement in physical function (-0.003 per year; 95% CI -0.005, -0.002; p<0.01). Mean and median values of pain and physical function, for each time point and per index joint and sub-sample, are reported in Table 3.

## Discussion

This is, to the best of our knowledge, the first study investigating the long-term benefits of patients adhering to a digital OA self-management programme. Results in two groups of 499 and 138 individuals being adherent to treatment for about 5 out of 7 days per week for 24 or 48 weeks showed a substantial reduction in the level of pain and an increase in physical performance per month of treatment. Based on available group-level data, there were no signs of

**Table 2. Descriptive data for knee and hip OA in the 48-week sub-sample (n = 138).**

| Characteristic | Knee (n = 78) | Hip (n = 60) |
|---|---|---|
| Age, mean (SD) | 65 (9) | 64 (8) |
| Female, n (%) | 54 (69) | 42 (70) |
| BMI, mean (SD) | 27.0 (4.4) | 26.0 (3.8) |
| **Weight status, n (%)** | | |
| • Underweight (BMI <18.5) | 0 (0) | 0 (0) |
| • Normal weight (BMI 18.5–24.9) | 29 (37) | 30 (50) |
| • Overweight (BMI 25–29.9) | 28 (36) | 17 (28) |
| • Obese (BMI ≥30) | 21 (27) | 13 (22) |
| **Baseline pain per severity group** | | |
| • Low, mean (SD) | 2.7 (1.0) | 3.8 (1.1) |
| • Moderate, mean (SD) | 5.5 (0.7) | 6.5 (0.7) |
| • Severe, mean (SD) | 7.6 (0.7) | 8.4 (0.7) |

BMI = body mass index. NRS = Numeric rating scale. OA = osteoarthritis.

worsening of symptoms during participation. Around 70% of those undertaking the programme reached a clinically relevant pain reduction at both follow-ups, and could thus be characterized as responders to treatment, with no clear difference between people with knee or hip OA. This result suggests that taking part in and adhering to a digital self-management programme is beneficial for reducing pain intensity and increasing physical function for older adults with knee or hip OA, and that results are maintained for up to one year.

In the present study, we observed greater improvements in both pain and physical function when compared to results from systematic reviews summarizing studies on people with knee and hip OA undergoing first-line interventions, where a decrease in pain of 6 points out of 100, at six months, was reported for knee OA [6, 7]. Yet these results are not directly comparable since the meta-analysis reported results several months after the close of active treatment, while the current study reports on a continuous intervention, but also due to potential differences in study population and intervention delivery.

The reduction in pain and increase in physical function in people with OA participating in first-line interventions has been shown to decline over time [6, 7, 18], probably due to the limited treatment duration (usually between 8 and 12 sessions over 8 weeks). Available evidence suggests that undertaking exercise for longer periods is associated with better outcomes [6, 30], regardless of the duration of a single session, potentially explaining the larger improvement experienced by people undergoing the present digital programme [31, 32].

The results of this study are comparable to, or somewhat better than shown in previous studies analysing the effect of digital self-management programmes. Bossen et al. showed a 2-point reduction on the WOMAC pain subscale (equivalent to a 10-point reduction on a 0–100 scale) after 8 months of providing a fully automated web-based programme aiming to increase patients' physical activity (n = 199) [21]. The intervention consisted of a 9-week gradually increasing program. The presence of targeted joint-specific exercises in the present digital self-management programme performed 5–10 minutes daily, combined with the inclusion of human supervision with continuous follow-up of goals and outcomes, may explain the greater pain reduction showed in the current study. Another self-management programme including education and exercise delivered digitally, showed a similar pain reduction after 6 months in comparison to the current study, although results were based on a small sample (n = 41), and the program ended after 12 weeks [22].

**Fig 4. Pain change over time, per index joint, for the 0 to 48-week subgroup of patients.** Symbols show mean pain value at 0, 12, 24 and 48 weeks for patient subgroups with index joint knee (n = 78) or hip (n = 60). Error bars represent 95% confidence intervals.

Participants in the present digital self-management programme with more severe pain experienced larger absolute pain reductions compared to those with less severe pain, yet the relative reduction was comparable. This suggests that the programme has the potential to yield similar benefit regardless of pain experienced by the participant at baseline. Hence it may be argued that the selection of patients for an exercise-based intervention should not be based solely on pain, and confirms that exercise is effective in reducing pain regardless of symptom severity [12].

Results from some exercise-based OA interventions suggest that knee OA patients experience a larger pain reduction, in comparison to those suffering from hip OA [18, 19]. We were unable to confirm this differential response to exercise of the knee and hip in the present study. Differences in participant' characteristics (e.g. age, disease severity), outcomes used, treatment delivery and exercise dose may explain these different study results.

Some limitations of our study need to be addressed. The lack of a control group and randomization limit our ability to explain the mechanisms of the observed reduction in pain and increased physical function in these OA patients, or to claim a direct cause and effect relationship. However, previous systematic reviews and meta-analyses of controlled trials provide evidence of a cause and effect relationship between exercise and patient benefits [33, 34].

In the current study pain and physical function were both shown to improve over time. Furthermore, patients of all pain levels at baseline improved (high variability in baseline pain was found). Even so, some contribution to outcomes by context effects and regression to the

**Table 3. Pain and physical function at baseline and follow-up.**

| Time | Mean (95% CI) | Median (IQR) | Mean (95% CI) | Median (IQR) |
|---|---|---|---|---|
| | **24-week sub-sample (n = 499)** | | | |
| | **Knee OA (n = 301)** | | **Hip OA (n = 198)** | |
| **Pain** | | | | |
| Baseline | 5.6 (5.4–5.8) | 6 (2) | 5.9 (5.7–6.2) | 6 (2) |
| Week 12 | 3.4 (3.2–3.7) | 3 (3) | 3.9 (3.7–4.3) | 4 (4) |
| Week 24 | 3.1 (2.9–3.4) | 3 (3) | 3.8 (3.4–4.1) | 3 (3) |
| **Physical function** | | | | |
| Baseline | 10.0 (9.6–10.4) | 10 (4) | 10.9 (10.4–11.5) | 10 (3) |
| Week 12 | 13.7 (13.2–14.2) | 13 (6) | 14.2 (13.4–14.9) | 13 (6) |
| Week 24 | 14.3 (13.7–14.8) | 14 (6) | 14.8 (13.9–15.5) | 14 (7) |
| | **48-week sub-sample (n = 138)** | | | |
| | **Knee OA (n = 78)** | | **Hip OA (n = 60)** | |
| **Pain** | | | | |
| Baseline | 5.7 (5.2–6.1) | 6 (3) | 5.8 (5.2–6.3) | 6 (3) |
| Week 12 | 3.2 (2.8–3.7) | 3 (3) | 3.9 (3.3–4.4) | 4 (4) |
| Week 24 | 2.9 (2.4–3.5) | 2 (3) | 3.4 (2.8–3.9) | 3 (3) |
| Week 48 | 3.2 (2.7–3.8) | 3 (4) | 3.8 (3.1–4.4) | 4 (4) |
| **Physical function** | | | | |
| Baseline | 10.3 (9.6–10.9) | 10 (4) | 11.1 (10.2–12.1) | 11 (4) |
| Week 12 | 13.7 (12.8–14.6) | 14 (5) | 14.6 (13.5–15.7) | 14 (6) |
| Week 24 | 15.1 (13.9–16.3) | 15 (6) | 15.4 (14.1–16.7) | 15 (7) |
| Week 48 | 14.4 (13.1–15.7) | 14 (6) | 14.9 (13.7–16.2) | 15 (5) |

[a]Measured using the Numerical Rating Scale (NRS 0–10).

mean would be expected, although the magnitude is uncertain. Only a randomized controlled trial can provide a specific answer to the degree of regression to the mean.

Since results are based on a register of patients voluntarily choosing whether to report their outcomes and when to end treatment, reflecting clinical reality, some data is missing. Although the results suggest that those ending treatment do not importantly differ from included participants in terms of descriptive factors at baseline, for future studies it would be of value to interview and follow those ending treatment, and their OA-related outcomes. Specifically, it would be valuable due to the current program being continuous and having no defined ending. The participant sample was drawn from a register and inclusion criteria comprised of a hip or knee OA diagnosis and having initiated treatment with a 70% adherence for the initial weeks, supporting generalisability. Hence, the sample should reflect the population to a greater extent than randomized controlled trials that commonly have more stringent inclusion criteria. Adding on, the 30 CST was instructed using a video and the patient thus performed the test accordingly without the supervision of a physiotherapist, hence the result from the measure should be interpreted with caution when compared to other studies. Finally, we could not control for other treatments undertaken by the participants during the study period, therefore use of concurrent non-pharmacological or pharmacological treatments may have influenced the results.

The challenges and barriers of delivering exercise and education based self-management programmes to the growing OA population are substantial. Considering the positive results showed in this and previous studies, digital interventions may represent a viable alternative for patients without access to or not interested in participating in traditional face to face

programmes. Digital interventions such as the present one may also complement traditional programmes to enhance long-term adherence to treatment. Yet further investigation into specific components of digital interventions outside of exercise is warranted, to more clearly understand the mechanics and benefits of specific parts of digital OA treatment programs.

## Supporting information

**S1 Table. Comparative data for the 24-week sub-sample (n = 499) and excluded participants (n = 1210).**
(PDF)

**S1 Checklist. STROBE statement—checklist of items that should be included in reports of** *cohort studies.*
(DOC)

## Acknowledgments

The authors would like to thank the participants of the study for contributing essential data. We express our gratitude to Vinnova, the Swedish Innovation Agency, for their financial support. Also, thanks to Jonas Ranstam for statistical guidance and advice, and to Daniel Kloek for graphical assistance.

## Author Contributions

**Conceptualization:** Leif E. Dahlberg, Andrea Dell'Isola, L. Stefan Lohmander, Håkan Nero.

**Data curation:** Håkan Nero.

**Formal analysis:** Håkan Nero.

**Funding acquisition:** Leif E. Dahlberg.

**Investigation:** Leif E. Dahlberg, Andrea Dell'Isola, L. Stefan Lohmander, Håkan Nero.

**Methodology:** Leif E. Dahlberg, Andrea Dell'Isola, L. Stefan Lohmander, Håkan Nero.

**Resources:** Leif E. Dahlberg.

**Visualization:** L. Stefan Lohmander.

**Writing – original draft:** Leif E. Dahlberg, Andrea Dell'Isola, L. Stefan Lohmander, Håkan Nero.

**Writing – review & editing:** Leif E. Dahlberg, Andrea Dell'Isola, L. Stefan Lohmander, Håkan Nero.

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
