## [Decision Letter · Decision Letter 0]

20 Dec 2019

PONE-D-19-29313

Improving osteoarthritis care by digital means - Effects of a digital self-management program after 24 weeks of adherence

PLOS ONE

Dear Dr. Nero,

Thank you for submitting your manuscript to PLOS ONE. After careful consideration, we feel that it has merit but does not fully meet PLOS ONE’s publication criteria as it currently stands. Therefore, we invite you to submit a revised version of the manuscript that addresses the points raised during the review process.

Please address the minor concerns of both reviewers when resubmitting your manuscript.

We would appreciate receiving your revised manuscript by Feb 03 2020 11:59PM. To enhance the reproducibility of your results, we recommend that if applicable you deposit your laboratory protocols in protocols.io, where a protocol can be assigned its own identifier (DOI) such that it can be cited independently in the future. For instructions see: http://journals.plos.org/plosone/s/submission-guidelines#loc-laboratory-protocols

We look forward to receiving your revised manuscript.

Kind regards,

Kelly Naugle, PhD

Academic Editor

PLOS ONE

Journal Requirements:

4. Thank you for providing the following Funding Statement: 

'Funding was received by LED from Vinnova - Sweden’s Innovation Agency (grant number: 2016-04187, www.vinnova.se) and Stiftelsen för Bistånd åt Rörelsehindrade i Skåne (grant number: 2019-01-20, www.stiftbistandskane.se) to the Department of Clinical Sciences Lund, Orthopaedics, Lund University, Sweden. In kind support (data gathering and extraction) was provided by Arthro Therapeutics Inc. The funders or supporter had no role in study design, data collection and analysis, decision to publish, or preparation of the manuscript.'

a. We note that one or more of the authors is affiliated with the funding organization, indicating the funder may have had some role in the design, data collection, analysis or preparation of your manuscript for publication; in other words, the funder played an indirect role through the participation of the co-authors.

If the funding organization did not play a role in the study design, data collection and analysis, decision to publish, or preparation of the manuscript and only provided financial support in the form of authors' salaries and/or research materials, please review your statements relating to the author contributions, and ensure you have specifically and accurately indicated the role(s) that these authors had in your study in the Author Contributions section of the online submission form. Please make any necessary amendments directly within this section of the online submission form.  Please also update your Funding Statement to include the following statement: “The funder provided support in the form of salaries for authors [insert relevant initials], but did not have any additional role in the study design, data collection and analysis, decision to publish, or preparation of the manuscript. The specific roles of these authors are articulated in the ‘author contributions’ section.”

If the funding organization did have an additional role, please state and explain that role within your Funding Statement.

Reviewers' comments:

Reviewer's Responses to Questions

**Comments to the Author**

1. Is the manuscript technically sound, and do the data support the conclusions?

Reviewer #1: Yes

Reviewer #2: Yes

2. Has the statistical analysis been performed appropriately and rigorously? 

Reviewer #1: Yes

Reviewer #2: Yes

3. Have the authors made all data underlying the findings in their manuscript fully available?

Reviewer #1: No

Reviewer #2: Yes

4. Is the manuscript presented in an intelligible fashion and written in standard English?

Reviewer #1: Yes

Reviewer #2: Yes

5. Review Comments to the Author

Reviewer #1: Thank you for the opportunity to review this longitudinal cohort study exploring the effects of a digital self-management program, and the impact of adherence at 6 and 12 months on outcomes. The study is sound in both methodology and reporting, however I would recommend a small number of minor considerations.

General comment/query – the use of the phrase ‘long-term’ is usually reserved for follow-up periods of 12 months or more?

Methods

Setting and Participants

‘About 95%’ – please make this a specific figure

E-questionnaire details – these would be better in the Outcome measures section

Adherence – was exercise completion logged by participants online? This is self-reported whereas the completion of quizzes/interaction with online content is objectively measured. Is this accounted for? Was there justification for including only participants who had adherence of 70% or more in the initial weeks?

Pain reports – was there justification for the choice of the window of +/- 4 or 6 weeks either side of the 24 and 48 weeks timepoints?

Outcome measures – joint pain - was this average pain over the past week?

Figures

Would benefit from formatting (currently directly taken from statistical package)

Discussion

The discussion focuses on comparison to other existing studies. A stronger case for the impact of the findings of this study would be beneficial.

Drop-outs between week 24 and week 48 are significant (n=400) – what are the implications of this for the results and also the programme moving forward (how might these participants be encouraged to remain in the programme).

Reviewer #2: General comments

This is a well-conducted registry-study, providing interesting insights in digital self-management of OA. The manuscript is clearly written and well-reported.

I agree that one of the great challenges in providing evidence-based first-line care for people with OA is findings ways to ensure both appropriate timing and order of treatments (exercise, weight management and education as first-line) as well ensuring treatment retention. Digital self-management may help support appropriate self-management for those people, who we fail to reach through traditional avenues. Digital self-management and better access to professional support (via chat or phone) used in conjunction with treatment strategies may also improve treatment retention. Thus, digital self-management may have several clinically important perspectives in securing optimal care for OA patients.

Apart from knowing very little about the effectiveness of non-surgical treatment options, regarding different exercise types and content of education and self-management, there is a big knowledge gap regarding the dosing of non-surgical treatment strategies for optimal effects.

Specific comments

Abstract:

Results and conclusion:

It is unclear to me, how big the effect on pain and function was. Is the reader supposed to sum up the monthly decrease in pain for 24 weeks and 48 weeks, respectively? I would prefer a more clear presentation of your findings at the two follow-ups.

“There were no clinically relevant effects on the improvement of pain or function

by any of the controlled factors” – what do you mean by controlled factors?

In the conclusion you change the time-period to over 6 months – which of the two follow-ups do you relate this to? Also, you state important decrease in pain and increased function. Do you mean clinically important? Above a certain threshold? Please clarify.

Methods:

Settings and participants

- Which diagnostic criteria did you use to confirm clinical OA via telephone?

- Please clarify the use of +/- 4 week intervals for 24-week follow-up and +/- 6 week intervals for 48-week follow-up. This seems a bit random and means that could essentially be up to 12 weeks difference when follow-up for 48-weeks was reported. This interval alone equals the length of most non-surgical treatment interventions. Please clarify.

- I cannot really understand the choice to divide subjects into the two subsamples (24-week and 48-week). Why not keep participants in one group? In relation to this, you do not really spend much time comparing the two groups anyway.

Outcome measures:

- The 30 s chair-stand test was originally developed as an objectively assessed measure of performance-based physical function. In this study, it was performed as a self-assessed measure. I would recommend extreme caution when interpreting findings from the test. Please discuss this in the context of the findings.

Results:

- The number of participants included in the 24-week subsample do not add up. 920 – 290 = 630. 630 – 138 = 492.

Discussion:

Most of the discussion concerns the exercise component of the digital self-management program. However, it seems unclear to me how much of the total program comprised of exercise (two daily exercises is not much) and how much comprised of self-management strategies. I would recommend a careful interpretation of exercise effects while also appreciating other elements of the program, which might have at least as important. A more in-depth discussion of the content of the program help the reader understand what was actually delivered.

- In relation to this, please clarify more specifically how this digital self-management program is different from other cited digital self-management programs.

- Please comment on the dosage of the program with regards to exercise and self-management compared to other digital and face-to-face programs. This will help clarify the clinical feasibility and sustainability of this intervention.

- Please comment on the clinical importance of the improvements in 30 s chair-stand.

- I would recommend the authors to refrain from discussing potential factors speaking against the potential involvement of regression to the mean. Patents typically seek care when their symptoms are elevated compared to their normal levels (this current sample). The fact that patients improve regardless of baseline pain levels does not speak against involvement of the regression of the mean phenomenon, since the pain levels are individually relative. Only randomization can provide a specific answer to the degree of regression to the mean.

- Please provide some perspectives on how we should move on from here, i.e. how do we explore working mechanisms of the program?

6. PLOS authors have the option to publish the peer review history of their article (what does this mean?). If published, this will include your full peer review and any attached files.

Reviewer #1: No

Reviewer #2: No

---

## [Author Response · Author response to Decision Letter 0]

3 Feb 2020

Addressing journal Requirements

Journal Requirements:

Action: Removed the capital E in the title.

Response: The digital program contains several questions and questionnaires at different time points, hence there is no one single questionnarie available. We will provide additional information regarding relevant questions and data.

Action: Added information on outcomes and questionnaires under Outcome measures, on page 7, at the beginning and the end of the paragraph.

Response: Data from this registry study contains potentially identifying and sensitive patient information. Since this is a registry study and we are unable to ask for consent from participants before sharing data (contact information of participants ending treatment is discarded, according to Swedish law), we limit the data sharing to be based upon a reasonable request via email.

Action: The above text has been added to the Data Sharing statement, page 16, to further clarify why we take extra precautions in terms of data.

Action: We have updated the cover letter.

4. Thank you for providing the following Funding Statement: 

'Funding was received by LED from Vinnova - Sweden’s Innovation Agency (grant number: 2016-04187, www.vinnova.se) and Stiftelsen för Bistånd åt Rörelsehindrade i Skåne (grant number: 2019-01-20, www.stiftbistandskane.se) to the Department of Clinical Sciences Lund, Orthopaedics, Lund University, Sweden. In kind support (data gathering and extraction) was provided by Arthro Therapeutics Inc. The funders or supporter had no role in study design, data collection and analysis, decision to publish, or preparation of the manuscript.'

a. We note that one or more of the authors is affiliated with the funding organization, indicating the funder may have had some role in the design, data collection, analysis or preparation of your manuscript for publication; in other words, the funder played an indirect role through the participation of the co-authors.

If the funding organization did not play a role in the study design, data collection and analysis, decision to publish, or preparation of the manuscript and only provided financial support in the form of authors' salaries and/or research materials, please review your statements relating to the author contributions, and ensure you have specifically and accurately indicated the role(s) that these authors had in your study in the Author Contributions section of the online submission form. Please make any necessary amendments directly within this section of the online submission form. Please also update your Funding Statement to include the following statement: “The funder provided support in the form of salaries for authors [insert relevant initials], but did not have any additional role in the study design, data collection and analysis, decision to publish, or preparation of the manuscript. The specific roles of these authors are articulated in the ‘author contributions’ section.”

If the funding organization did have an additional role, please state and explain that role within your Funding Statement.

Action: We have added the recommended statement to the Financial Disclosure Statement, please see page 3 in the manuscript.

...” The funder provided support in the form of consulting fees for authors [LED, LSL, HN], but did not have any additional role in the study design, data collection and analysis, decision to publish, or preparation of the manuscript. The specific roles of these authors are articulated in the ‘author contributions’ section.”.. 

Action: The Competing Interests statement has been updated accordingly, please see page 15 in the manuscript.

Action: The cover letter has been updated accordingly. 

Rebuttal letter PONE-D-19-29313

To the editor and reviewers,

Thank You for the opportunity to improve our manuscript Improving osteoarthritis care by digital means - effects of a digital self-management program after 24 weeks of adherence, referenced PONE-D-19-29313. We have addressed the comments by the reviewers and hope that our revision will be satisfactory. Please find below our point-by-point responses and actions to each of the editors and reviewers’ concerns. The reviewer comments are written in italics and changes are referred to in the manuscript based on page number and section as well as quoted after each described action.

Reviewer #1: Thank you for the opportunity to review this longitudinal cohort study exploring the effects of a digital self-management program, and the impact of adherence at 6 and 12 months on outcomes. The study is sound in both methodology and reporting, however I would recommend a small number of minor considerations.

1. General comment/query – the use of the phrase ‘long-term’ is usually reserved for follow-up periods of 12 months or more?

Response: Thank you for the taking the time to review our manuscript. We used the phrase long-term since the most common treatment period in terms of first-line treatment for osteoarthritis (OA) is from 6 weeks to 3 months, and to the best of our knowledge no other study has reported results from continuous treatment for 6 months or more.

Action: None.

2. Methods

Setting and Participants

‘About 95%’ – please make this a specific figure

Response: Thank you for the suggestion, we will specify it as 95%, as this was the percentage at the time of data extraction.

Action: Removed the word “About” before “95%” below Setting and participants, page 5.

..”Included participants had a radiographic and or clinical diagnosis of hip or knee OA from a physical therapist or physician (95% of all patients at the date of data extraction).”…

3. E-questionnaire details – these would be better in the Outcome measures section

Response: Thanks, the section has been moved accordingly, and also altered to clarify details on outcomes (as suggested by the editor).

Action: Moved text from under Setting and participants page 5, to Outcome measures, page 7 in the manuscript.

..”Data on the participants’ overall health and characteristics (age, gender, BMI) as well as OA-related factors (most painful joint, afflicted side) were collected at inclusion.”…

4. Adherence – was exercise completion logged by participants online? This is self-reported whereas the completion of quizzes/interaction with online content is objectively measured. Is this accounted for? Was there justification for including only participants who had adherence of 70% or more in the initial weeks?

Response: Exercise completion was indicated by patient actions through the smartphone app. When exercises (in video format with explanatory graphics) were distributed to the patient, the patient was asked to rate the perceived difficulty of the exercise. Entering the perceived difficulty was used as indicator that the exercise was performed. Completion of quizzes/lessons was indicated by the patient answering the quizzes correctly. Program continuation was halted until these tasks were completed. Agreed, there is some difference in terms of how lessons and exercises are handled, and we have made a slight change in the manuscript to clarify this.

We intended to monitor the effect of participation in the digital treatment, and arbitrarily chose the 70% cut-off as a threshold for what could reasonably be considered participation in treatment. 

Action: Information on collection of information on completed exercises and lessons has been added to the Intervention section, please see page 6 and 7 of the manuscript. 

…“While rating perceived difficulty and adding comments, patients also indicated when exercises were completed. Program continuation was halted until all exercises for the day were marked as completed.”…

…“Completion of a lesson was indicated by the patient answering the quiz correctly.”…

5. Pain reports – was there justification for the choice of the window of +/- 4 or 6 weeks either side of the 24 and 48 weeks timepoints?

Response: The weekly reporting of pain is not mandatory, to alleviate patient strain and add more flexibility. Around 54% of the 24 week sub-sample had reported pain at week 24. Similarly, around 50% of participants active at week 48 had reported pain this specific week. Hence we opted for the +/- 4 weeks and +/- 6 weeks window to collect the maximum amount of data. We will clarify this in the manuscript.

Action: Clarified information on time windows on page 6, Setting and participants:

..”Outcome analysis was made in two separate sub-samples; participants with a pain report from week 24 or adjacent week (+- 4 weeks) and week 48 or adjacent week (+- 6 weeks), respectively. Adjacent weeks were added to collect the maximum amount of data (around 50% of active participants had reported pain at week 24 and 48, respectively).”..

6. Outcome measures – joint pain - was this average pain over the past week?

Response: Yes, as stated under Outcome measures: ..”Mark on this scale how much pain you had the last week in your hip/knee”…

Action: None.

7. Figures

Would benefit from formatting (currently directly taken from statistical package)

Response: Thank you for the suggestion, the figures have been formatted/updated. Please see new figure 2, 3 and 4 below.

Action: Formatted figures and updated the manuscript files, also attached new figures below:

8. Discussion

The discussion focuses on comparison to other existing studies. A stronger case for the impact of the findings of this study would be beneficial.

Drop-outs between week 24 and week 48 are significant (n=400) – what are the implications of this for the results and also the programme moving forward (how might these participants be encouraged to remain in the programme).

Response: Thanks for highlighting these issues. We would like to offer an alternative view on the matter. We believe that, in the current scientific community, you must argue for the impact of your study by comparing it to the work of other researchers. If no comparison is available, an evaluation is hard. We also believe that we have in fact argued for the impact of our findings, by first comparing to previous work and then highlighting our unique findings, or our confirmatory findings, and their potential impact.

For example, from page 14, second paragraph: ...” Participants in the present digital self-management programme with more severe pain experienced larger absolute pain reductions compared to those with less severe pain, yet the relative reduction was comparable. This suggests that the programme has the potential to yield similar benefit regardless of pain experienced by the participant at baseline. Hence it may be argued that the selection of patients for an exercise-based intervention should not be based solely on pain, and confirms that exercise is effective in reducing pain regardless of symptom severity [12].”...

In regard to drop-outs, it is more complicated to define a drop-out when you administer a program that has no defined ending. The treatment ends when the patient considers it appropriate, or when the physiotherapist deems it suitable. Our previous qualitative studies indicated several reasons for the patient quitting the program, such as: they had improved to an extent where continued treatment seemed unnecessary; they had not improved sufficiently and was seeking other options; or that the patient continued exercising on their own without the smartphone application (1). Interestingly, the first pilot study based on this program showed no difference between those ending treatment and those continuing, in terms of level of pain (2). Unfortunately for the current study we do not have data (on a group level) on what was the specific reason for patients to quit the program. Adding on, patients in the register have started the treatment sequentially, hence not all participants in our sample have had the opportunity to be in the program for 48 weeks. In fact, only 710 individuals of the 1709 had started more than or equal to 48 weeks before data extraction.

Since OA is a chronic disease the treatment needs to be continuous yet changing life-style routines and habits in patients is, as we know, hard. Research on our end is ongoing, trying to understand when and why patients stop adhering, and we are also interested in scientifically testing some different behavioral strategies to prolong adherence.

To further clarify the matter of drop-outs, we have edited some text in the Method section, added some information to the topic in the Discussion and edited Figure 1.

Action: Added/edited the following text in the Discussion, page 15:

..” Since results are based on a register of patients voluntarily choosing whether to report their outcomes and when to end treatment, reflecting clinical reality, some data is missing. Although the results suggest that those ending treatment do not importantly differ from included participants in terms of descriptive factors at baseline, for future studies it would be of value to interview and follow those ending treatment, and their OA-related outcomes. Specifically, it would be valuable due to the current program being continuous and having no defined ending.”...

Added the following text to the Method section, page 6:

..”At that time point, the register contained data from 1709 Swedish participants (710 individuals had started ≥48 weeks before data extraction) that had reported one of their knees or hips as their most symptomatic joint (index joint), had been treated in the programme for at least three weeks with a minimum adherence of 70% and had registered ≥24 weeks before data extraction.”..

Edited Figure 1 to clarify how many had started the program 48 weeks before data extraction, and how many were excluded from the 24-week sub-sample due to data at week 48 or adjacent:

 

Reviewer #2: General comments

1. This is a well-conducted registry-study, providing interesting insights in digital self-management of OA. The manuscript is clearly written and well-reported.

I agree that one of the great challenges in providing evidence-based first-line care for people with OA is findings ways to ensure both appropriate timing and order of treatments (exercise, weight management and education as first-line) as well ensuring treatment retention. Digital self-management may help support appropriate self-management for those people, who we fail to reach through traditional avenues. Digital self-management and better access to professional support (via chat or phone) used in conjunction with treatment strategies may also improve treatment retention. Thus, digital self-management may have several clinically important perspectives in securing optimal care for OA patients.

Apart from knowing very little about the effectiveness of non-surgical treatment options, regarding different exercise types and content of education and self-management, there is a big knowledge gap regarding the dosing of non-surgical treatment strategies for optimal effects.

Response: Thank you for your help in improving our manuscript. Agreed, there are unknown aspects about optimal dosage or dose-response in terms of first-line treatment for osteoarthritis. What we do know is that most programs reporting beneficial results contain a minimum of 3 sessions per week, focus on single-type exercises (e.g. strengthening muscles surrounding the painful joint), and are supervised (3). Our study adds to the evidence in terms of duration, showing that treatment for a minimum of 6 months leads to substantial improvements in terms of pain.

Action: None.

Specific comments

Abstract:

2. Results and conclusion:

It is unclear to me, how big the effect on pain and function was. Is the reader supposed to sum up the monthly decrease in pain for 24 weeks and 48 weeks, respectively? I would prefer a more clear presentation of your findings at the two follow-ups.

Response: We were interested in reporting the effect per month of duration to add to the evidence of exercise duration and its effects on pain and function. We felt this adds more valuable information than just a mean difference, seeing as the data underlying the calculation was weekly reports. Adding on, means for baseline and follow-ups can be found in table 3. However, we concur that the results in the abstract can be clearer in this aspect and will revise the abstract accordingly.

Action: Mean numbers for baseline and follow-up was added to the Methods and Findings section of the abstract, and the results were further clarified:

..” For the 24-week sub-sample, pain NRS decreased monthly by -0·43 units (95% CI -0·51, -0·35, mean knee pain from 5.6 to 3.1, and hip pain from 5.9 to 3.8) and 30CST repetitions increased monthly by 0·76 repetitions (95% CI 0·64, 0·89 mean for knee from 10.0 to 14.3, and for hip from 10.9 to 14.8). For the 48-week sub-sample, pain decreased monthly by -0·39 units (95% CI -0·43, -0·36, mean knee pain from 5.7 to 3.2, and hip pain from 5.8 to 3.8), and repetitions increased by 0·72 repetitions (95% CI 0·65, 0·79, mean repetitions for knee from 10.3 to 14.4, and for hip from 11.1 to 14.9).”...

3. “There were no clinically relevant effects on the improvement of pain or function

by any of the controlled factors” – what do you mean by controlled factors?

Response: Thanks for highlighting this, in fact the proper term should be covariates, as described in the Statistical analysis. This needs to be clarified.

Action: Added text in the Abstract – Methods and findings:

..” There were no clinically relevant effects on the improvement of pain or function by any covariate (age, sex, index joint).”...

4. In the conclusion you change the time-period to over 6 months – which of the two follow-ups do you relate this to? Also, you state important decrease in pain and increased function. Do you mean clinically important? Above a certain threshold? Please clarify.

Response: The word over may confuse hence it will be changed to for. In terms of important increase, we refer to the fact that it was a clinically relevant improvement, and as referred to in the manuscript, on page 7 under Outcome measures, above what was previously defined as minimal clinically relevant change (20%) for pain. We will clarify this.

Action: Changed the text in the Conclusion of the Abstract, page 3:

..” Continuously participating in a digital OA treatment program for 6 months was associated with a clinically important decrease in joint pain and increased physical function, in hip and knee OA.”...

Methods:

Settings and participants

5. - Which diagnostic criteria did you use to confirm clinical OA via telephone?

Response: 

The NICE guidelines were used as diagnostic criteria, also acknowledged in the Swedish National Guidelines, stating that OA can be diagnosed without further evaluation if the person is 45 years old or over, has activity-related joint pain, has either no morning stiffness or morning stiffness that lasts no longer than 30 minutes. It was also confirmed that no atypical features, or red flags, were present.

Information relating to symptoms is available to the physiotherapist before the initial telephone meeting. The physiotherapist is obliged during the telephone call to confirm all collected data and ask clinically relevant questions to rule out differential diagnosis. After including more than 12.000 patients into the program, there has been no problem related to improper diagnosing or adverse events.

We have added a small paragraph under Methods, Setting and participants, page 5 to clarify.

Action: Added some information on red flags under Settings and participants, page 5:

..”Individuals without a prior diagnosis had clinical OA confirmed by an orthopaedic surgeon or physiotherapist via telephone (diagnosis according to NICE criteria and Swedish National Guidelines, and confirming the absence of any red flag symptoms), or if deemed necessary were recommended to seek face-to-face care before inclusion in the programme.”..

6. - Please clarify the use of +/- 4 week intervals for 24-week follow-up and +/- 6 week intervals for 48-week follow-up. This seems a bit random and means that could essentially be up to 12 weeks difference when follow-up for 48-weeks was reported. This interval alone equals the length of most non-surgical treatment interventions. Please clarify.

Response: As stated in the response to Reviewer 1, question number 5, the weekly reporting of pain is not mandatory, to alleviate patient strain and add more flexibility. After inspecting the data, around 54% of the 24 week sub-sample had reported pain at week 24. Similarly, around 50% of participants active at week 48 had reported pain this specific week. Hence we opted for the +/- 4 weeks and +/- 6 weeks window to collect the maximum amount of data.

Action: None.

7. - I cannot really understand the choice to divide subjects into the two subsamples (24-week and 48-week). Why not keep participants in one group? In relation to this, you do not really spend much time comparing the two groups anyway.

Response: If we would merge the 2 groups, we would increase the number up to 24w, but would have a larger dropout after this, and the patients responding at 48w would not be the same as those responding at 24w. We were interested in reporting results of prospective longitudinal cohorts with minimum loss to follow-up. Adding on, in a Swedish face-to-face program with published data from its registry, there is a worsening of symptoms (pain e.g.) between 3-12 months (4), hence it was interesting for us to see if this was also evident for a digital continuous program, and if so when the deterioration occurred. Hence we wanted separated cohorts in terms of duration of adherence. We clarified this in the title of the manuscript.

Action: Edited the title on page 1:

..”Improving osteoarthritis care by digital means - effects of a digital self-management program after 24- or 48-weeks of treatment.”..

8. Outcome measures:

- The 30 s chair-stand test was originally developed as an objectively assessed measure of performance-based physical function. In this study, it was performed as a self-assessed measure. I would recommend extreme caution when interpreting findings from the test. Please discuss this in the context of the findings.

Response: The performance of the test is pedagogically instructed using a video with text that can be played as many times as needed. The instructions are according to those published by the Swedish Physiotherapy Association. Hence the instructions are the same that a physiotherapist interprets in the clinic, where there may also be individual differences. Furthermore, the test and its result are used intra-person, hence we do not use the result to compare with other studies or individuals. Following your comment, we have added a paragraph on the interpretation of the test in the Discussion.

Action: Added a paragraph in the Discussion, page 15:

..”Adding on, the 30 CST was instructed using a video and the patient thus performed the test accordingly without the supervision of a physiotherapist, hence the result from the measure should be interpreted with caution.”..

9. Results:

- The number of participants included in the 24-week subsample do not add up. 920 – 290 = 630. 630 – 138 = 492.

Response: Following the flow chart in figure 1, it does add up. Most likely, from your calculation above, the 7 participants missing in your calculation are individuals with 48 week data that were missing data at week 24. Hence counting backwards, it does not seem to add up. We clarified this in the Results, thank you for pointing this out.

Action: Added information in the Results, page 9:

..” For the 48-week sub-sample, a total of 138 individuals (n=7 missing 24-week data included) with pain data at 48 weeks were included (Figure 1).”...

10. Discussion:

Most of the discussion concerns the exercise component of the digital self-management program. However, it seems unclear to me how much of the total program comprised of exercise (two daily exercises is not much) and how much comprised of self-management strategies. I would recommend a careful interpretation of exercise effects while also appreciating other elements of the program, which might have at least as important. A more in-depth discussion of the content of the program help the reader understand what was actually delivered.

Response: Reading through the discussion once again, we do not completely agree that most of the discussion concern the exercise component only. Most of the discussion, we suggest, concerns program specific components, one of them being exercise. There is one paragraph on page 14 that focuses primarily on exercise, this due to the fact that there is an abundance of studies investigating the effects of exercise, as compared to the other components such as education, physiotherapy support, automatic notifications, etc. It would be highly interesting to investigate specific effects of these individual components, but that would be out of scope for this study. If self-management is interpreted as theoretical education about the disease and how to self-manage in daily life, we know from previous studies based on face-to-face self-management programs, that the theoretical part of the program (patient education) alone was associated with less improvement of symptoms than education combined with exercise (4). Whether this applies to digital programs as well, is still unknown.

We have added some text in the Discussion concerning the content of the program. 

Action: Added text to the Discussion, page 16:

..” Yet further investigation into specific components of digital interventions outside of exercise are warranted, to more clearly understand the mechanics and benefits of specific parts of digital OA treatment programs.”...

11. - In relation to this, please clarify more specifically how this digital self-management program is different from other cited digital self-management programs.

Response: To the best of our understanding, the unique aspects of Joint Academy are that each patient is connected to a dedicated personal physiotherapist which the patient can contact at any time of day for advice or discussion. Furthermore, the program necessitates no added equipment to perform and start exercising. Thirdly, according to our knowledge, this is the only digital self-management program that is based on a nationally implemented face-to-face initiative that has been delivered to more than 120.000 individuals in Sweden.

The study from Bossen et al cited in the manuscript describes a web-based intervention focusing on increasing patient-selected physical activity, in increments. The intervention lasts for 9 weeks and for the study n=199 individuals were recruited. The other cited study describing a digital program, by Smittenaar et al, is based on n=41 individuals participating in a 12 week program. Participation necessitates wearing several sensors while exercising. 

Action: Added some info on the two cited studies on digital programs, in the Discussion, page 14:

..”Bossen et al. showed a 2-point reduction on the WOMAC pain subscale (equivalent to a 10-point reduction on a 0-100 scale) after 8 months of providing a fully automated web-based programme aiming to increase patients’ physical activity (n=199) [21]. The intervention consisted of a 9-week gradually increasing program.”..

..”Another self-management programme including education and exercise delivered digitally, showed a similar pain reduction after 6 months in comparison to the current study, although results were based on a small sample (n=41), and the program ended after 12 weeks [22].”..

12. - Please comment on the dosage of the program with regards to exercise and self-management compared to other digital and face-to-face programs. This will help clarify the clinical feasibility and sustainability of this intervention.

Response: It is hard to access or even find other digital programs similar to Joint Academy offering treatment for osteoarthritis of the hip or knee. As previously noted in the response to Question 11, we have referenced to two other programs focusing on knee pain or OA. There is also a digital program called Kaia that distributes exercises every day at about the same amount as Joint Academy but focusing on unspecified lower back pain. In terms of face-to-face programs, due to the effort of frequent visits to a clinic, most programs offer exercise sessions a few times per week, but with longer session duration. These programs also have a specified duration (3 months for BOA and the GLA:D programs) and thereby data on how much exercise is performed after the program may be sparse. We hypothesize, as mentioned in the Discussion on page 13-14, that the greater mean improvements reported in the current study is due to the constant availability of support from a physiotherapist, and the high frequency of training.

Action: Added a few words on high frequency of exercises, in the Discussion, page 14:

..”The presence of targeted joint-specific exercises in the present digital self-management programme performed 5-10 minutes daily, combined with the inclusion of human supervision with continuous follow-up of goals and outcomes, may explain the greater pain reduction showed in the current study.”..

13. - Please comment on the clinical importance of the improvements in 30 s chair-stand.

Response: There are published thresholds for what constitutes a major clinically important improvement in people with hip osteoarthritis performing the 30 CST (5). These vary between an increase of 2.0 - 2.6 repetitions, depending on method. Hence our result of an increase of 3.8 – to 3.9 (depending on sub-sample) for the hip OA patients would constitute a clinically important improvement.

Action: None.

14. - I would recommend the authors to refrain from discussing potential factors speaking against the potential involvement of regression to the mean. Patents typically seek care when their symptoms are elevated compared to their normal levels (this current sample). The fact that patients improve regardless of baseline pain levels does not speak against involvement of the regression of the mean phenomenon, since the pain levels are individually relative. Only randomization can provide a specific answer to the degree of regression to the mean.

Response: Thanks for highlighting this issue. OA can be intermittent in regard to symptoms such as pain, so a variation over time would not be surprising. The variation in pain at baseline for the different severity groups is; low: 0-5 (minimum-maximum), moderate: 5-7 (min-max) and severe: 7-10 (min-max). This would indicate quite a spread in terms of pain at baseline, although not disproving the effect of regression to the mean it indicates that perhaps not all patients seek treatment mainly due to severe pain. In fact, it has been suggested that care seeking behavior seems to be based on a mix of social and psychological factors and not simply based on the absence or presence of medical problems (6). Whether care seeking behavior in relation to digital programs differs from face-to-face treatment, is unknown. For certain societal groups and diagnoses digital care may be more accessible and hence care seeking may be related to other factors.

Furthermore, regression to the mean typically constitutes a value closer to the mean following an extreme measure, and as mentioned we cannot disprove this effect in the current study. However, we can argue that the effect of regression to the mean is reduced when taking serial measurements and calculating the average change (7). 

Yet after a brief discussion of this phenomena, we agree that randomization in a controlled study is what can provide a specific answer to the degree of the regression to the mean, and we will edit the manuscript to this effect.

Action: Added and edited a paragraph in the Discussion, page 15:

..” In the current study pain and physical function were both shown to improve over time. Furthermore, patients of all pain levels at baseline improved (high variability in baseline pain was found). Even so, some contribution to outcomes by context effects and regression to the mean would be expected, although the magnitude is uncertain. Only a randomized controlled trial can provide a specific answer to the degree of regression to the mean.”…

15. - Please provide some perspectives on how we should move on from here, i.e. how do we explore working mechanisms of the program?

Response: With digital programs and the possibility of collecting vast amounts of data on a scale not possible with face-to-face programs, the option of using machine learning algorithms to test and examine what mechanisms are most important for specific sub-groups is available. In a digital setting it is possible to rather easily alter parts of the program to a randomized set of participants, and this is what we are hoping to achieve in the near future.

We have added a comment in the discussion on this subject as mentioned in the response to question number 10 above, but not specified further, as we feel it is outside of the scope of this study.

Action: On page 16 in the Discussion, some text was added:

..” Yet further investigation into specific components of digital interventions outside of exercise are warranted, to more clearly understand the mechanics and benefits of specific parts of digital OA treatment programs.”...

 

References

1. Cronström A et al. 'I Would Never Have Done It if It Hadn't Been Digital': A Qualitative Study on Patients' Experiences of a Digital Management Programme for Hip and Knee Osteoarthritis in Sweden. BMJ Open, 9 (5), e028388. 2019.

2. Dahlberg LE et al. A Web-Based Platform for Patients With Osteoarthritis of the Hip

and Knee:A Pilot Study. JMIR Res Protoc. 5 (2), e115. 2016.

3. Juhl C et al. Impact of Exercise Type and Dose on Pain and Disability in Knee Osteoarthritis. Arthritis Rheumatol, 66 (3), 622-36. 2014.

4. Jönsson T et al. The Better Management of Patients With Osteoarthritis Program: Outcomes After Evidence-Based Education and Exercise Delivered Nationwide in Sweden. PLoS One. 14 (9), e0222657. 2019.

5. Wright AA et al. A Comparison of 3 Methodological Approaches to Defining Major Clinically Important Improvement of 4 Performance Measures in Patients With Hip Osteoarthritis. J Orthop Sports Phys Ther. 41 (5), 319-27. 2011.

6. Campbell SM et al. Why Do People Consult the Doctor? Fam Pract. 13 (1), 75-83. 1996.

7. Morton V et al. Effect of regression to the mean on decision making in health care. BMJ;326:1083. 2003.

---

## [Decision Letter · Decision Letter 1]

14 Feb 2020

Improving osteoarthritis care by digital means - effects of a digital self-management program after 24- or 48-weeks of treatment

PONE-D-19-29313R1

Dear Dr. Nero,

We are pleased to inform you that your manuscript has been judged scientifically suitable for publication and will be formally accepted for publication once it complies with all outstanding technical requirements.

With kind regards,

Kelly Naugle, PhD

Academic Editor

PLOS ONE

Additional Editor Comments (optional):

Reviewers' comments:

Reviewer's Responses to Questions

**Comments to the Author**

1. If the authors have adequately addressed your comments raised in a previous round of review and you feel that this manuscript is now acceptable for publication, you may indicate that here to bypass the “Comments to the Author” section, enter your conflict of interest statement in the “Confidential to Editor” section, and submit your "Accept" recommendation.

Reviewer #2: All comments have been addressed

2. Is the manuscript technically sound, and do the data support the conclusions?

Reviewer #2: Yes

3. Has the statistical analysis been performed appropriately and rigorously? 

Reviewer #2: Yes

4. Have the authors made all data underlying the findings in their manuscript fully available?

Reviewer #2: No

5. Is the manuscript presented in an intelligible fashion and written in standard English?

Reviewer #2: Yes

6. Review Comments to the Author

Reviewer #2: I appreciate the authors efforts in accommodating the manuscript according to comments and concerns raised. I have no further concerns regarding the reporting of this study.

I look forward to learn more about digital self-management in the treatment of OA in future work from your end.

7. PLOS authors have the option to publish the peer review history of their article (what does this mean?). If published, this will include your full peer review and any attached files.

Reviewer #2: No

---

## [Editor Report · Acceptance letter]

19 Feb 2020

PONE-D-19-29313R1 

Improving osteoarthritis care by digital means - effects of a digital self-management program after 24- or 48-weeks of treatment 

Dear Dr. Nero:

I am pleased to inform you that your manuscript has been deemed suitable for publication in PLOS ONE. Congratulations! Your manuscript is now with our production department. 

With kind regards,

on behalf of

Dr. Kelly Naugle 

Academic Editor

PLOS ONE